# Psychological Distress among Bangladeshi Dental Students during the COVID-19 Pandemic

**DOI:** 10.3390/ijerph19010176

**Published:** 2021-12-24

**Authors:** Farah Sabrina, Mohammad Tawfique Hossain Chowdhury, Sujan Kanti Nath, Ashik Abdullah Imon, S. M. Abdul Quader, Md. Shahed Jahan, Ashek Elahi Noor, Clopa Pina Podder, Unisha Gainju, Rina Niroula, Muhammad Aziz Rahman

**Affiliations:** 1Department of Oral & Maxillofacial Surgery, Update Dental College, Dhaka 1711, Bangladesh; sabrinafarahbds40@gmail.com (F.S.); imonomfs@gmail.com (A.A.I.); 2Department of Dental Public Health, Sapporo Dental College, Dhaka 1230, Bangladesh; knsujan@yahoo.com (S.K.N.); rajet.elahi@gmail.com (A.E.N.); drcppodder@gmail.com (C.P.P.); 3Department of Conservative Dentistry & Endodontics, Update Dental College, Dhaka 1711, Bangladesh; smilezonedental@yahoo.com; 4Department of Dental Public Health, Update Dental College, Dhaka 1711, Bangladesh; shahed.jahan5@gmail.com; 5Update Dental College, Dhaka 1711, Bangladesh; unesa97@gmail.com (U.G.); niroularina@gmail.com (R.N.); 6School of Health, Federation University Australia, Berwick 3806, Australia; 7Department of Non-communicable Diseases, Bangladesh University of Health Sciences (BUHS), Dhaka 1216, Bangladesh; 8Faculty of Public Health, Universitas Airlangga, Surabaya 60115, Indonesia

**Keywords:** psychological distress, fear, coping, COVID-19, Bangladesh, dental, mental health

## Abstract

*Background:* Psychological sufferings are observed among dental students during their academic years, which had been intensified during the COVID-19 pandemic. *Objectives:* This study assessed the levels and identified factors associated with psychological distress, fear and coping experienced by dental undergraduate students in Bangladesh. *Methods:* A cross sectional online survey was conducted during October-November, 2021. The Kessler Psychological Distress Scale (K-10), Fear of COVID-19 Scale (FCV-19S) and Brief Resilient Coping Scale (BRCS) were used in order to assess psychological distress, fear and coping strategies, respectively. *Results:* A total of 327 students participated; the majority (72%) were 19–23 years old and females (75%). One in five participants were infected with COVID-19 and 15% reported contact with COVID-19 cases. Negative financial impact (AOR 3.72, 95% CIs 1.28–10.8), recent or past COVID-19 infection, and contact with COVID-19 cases were associated with higher levels of psychological distress; but being a third year student (0.14, 0.04–0.55) and being satisfied about current social life (0.11, 0.03–0.33) were associated with lower levels of psychological distress. Being a third year (0.17, 0.08–0.39) and a fourth year student (0.29, 0.12–0.71) were associated with lower levels of fear. Health care service use and feeling positive about life were associated with medium to high resilience coping. *Conclusions:* This study identified dental students in Bangladesh who were at higher risk of psychological distress, fear and coping during the ongoing pandemic. Development of a mental health support system within dental institutions should be considered in addition to the academic and clinical teaching.

## 1. Introduction

The ongoing coronavirus disease (COVID-19) pandemic has been linked to more than 140 million cases worldwide, with approximately 3 million deaths [1]. The pandemic has caused the most cases and deaths in the United States of America, followed by India, Brazil, France, the Russian Federation, the United Kingdom, Turkey, Italy, and Spain. The first three cases of COVID-19 were identified in Bangladesh on 8 March 2020. As of 29 November 2021, Bangladesh has reported 1,575,579 confirmed COVID-19 cases and 27,975 deaths [2]. In response to the pandemic crisis, the Government of the People’s Republic of Bangladesh has designed a Multisectoral Action Plan. Lockdown in major cities, social distancing, closure of schools and universities, working from home arrangements where possible, widespread public awareness campaigns for handwashing practices, use of masks in public places, imposed regulations on international travel from hotspots, management of quarantine centers and nationwide testing facilities were a few of the initiatives taken by the Government to mitigate the impact of the pandemic. In addition, guidelines for COVID-19 clinical management were developed, public and private hospitals were designated for treating positive cases, isolation units were established in various hospitals, and regular public reporting was initiated based on the surveillance of COVID-19 cases and deaths [3,4].

The pandemic has impacted global communities in different ways. Besides the impact on physical health, it also triggered a slew of psychological issues, including panic disorder, anxiety, and sadness, in both COVID-19 patients and healthy people [5]. Due to the contagious nature of COVID-19, concerns such as spatial segregation, lockdown, travel limitations, and isolation, as well as social and economic ramifications, resulting in despair, anxiety, fear, panic, stress, suicidal thoughts, post-traumatic stress disorder and other mental health issues [6]. A recent study examining factors associated with psychological distress, fear of COVID-19, and coping across diverse community members in 17 countries showed that doctors had greater levels of psychological distress but lower levels of fear of COVID-19, whereas nurses had higher levels of resilient coping. Females and individuals with pre-existing mental health issues were identified as the most vulnerable groups of people having COVID-related psychological impact [7]. Bangladeshi individuals also experienced a great deal of psychological discomfort and terror, according to a recent study [8]. People with pre-existing mental health problems, females, frontline workers or essential service workers, current and one-time smokers, providing care to a known or suspected case of COVID-19, having an overseas travel history, being in quarantine, having positive test results for COVID-19, and having higher levels of fear of COVID-19 were associated with higher psychological distress [8]. The study also demonstrated that having an income source was associated with medium-to-high resilient coping [8]. The relationship between stress and coping had been explained by Lazarus and Folkman; stress could be explained by primary and secondary appraisals of the situation, whereas coping could be emotion-focused or problem-focused. Depending on the way people respond to a stressful situation, they could demonstrate adaptive or maladaptive coping behaviours [9].

The pandemic has posed a challenge to healthcare workers including dental clinicians around the world, prompting a variety of responses. Medical and Dental schooling are widely seen as demanding environments, with students experiencing higher levels of stress, anxiety, and depression than classmates studying other subjects [10]. Obtaining a Bachelor’s degree in Dentistry is a time-consuming process that demands extensive study and expertise of the discipline. In Bangladesh, the program is of five years duration, with the last two years dedicated to clinical training and a yearlong internship following graduation. An undergraduate Dentistry student needs to demonstrate theoretical knowledge, practical experience, and interpersonal skills, all of which are assessed at the end of each academic year via oral, written, and practical assessments. These ongoing academic responsibilities, as well as non-academic stress such as coordinating with faculty and administrative formalities, are often overwhelming for students [11,12]. Most dental treatments, particularly those involving the use of a dental hand piece, produce aerosols. During the pandemic, that practice potentially increased the risk of spreading infection at the dentist practice. It has been demonstrated that the virus in the aerosol may survive for more than 3 h, with surface stability over 72 h [13]. Development of that scientific evidence generated anxiety and stress amongst the students doing their clinical placement. Many countries have postponed elective dental procedures, and a few countries have even closed dental schools, clinics and teaching hospitals [14,15]. For months, dental clinics, dental schools, dental teaching hospitals and universities were closed in many countries such as USA, Canada, Japan, China, India etc. [16,17,18]. In addition, all academic dental institutions and dental clinics in Bangladesh were temporarily closed during the pandemic. Only emergency dental treatments were provided.

Following the necessity for social distancing due to the COVID-19 pandemic, physical presence at schools, colleges, and universities around the world was restricted and transitioned to a virtual learning environment. There were similar arrangements for Dental schools all over the world including Bangladesh. Such a new way of learning allowed the academics and students to gain more personalized educational experience. In addition, for preclinical simulation exercises, certain teaching institutes adopted the social distancing methods in their dental laboratories following the strict COVID-19 guidelines (for example: students were divided into small subgroups in their clinical class, wearing masks and face shields, using hand sanitizers) [19,20,21]. However, evidence showed that Dental and Medical students suffered from psychological anguish due to the change in learning environment during their academic and professional years [22]. Prior evidence showed that dental students reported a number of mental health issues including depression, anxiety, obsessive-compulsive disorders and interpersonal sensitivity in their academic years during the pre-pandemic environment [23,24,25,26]. However, very limited evidence was generated from South Asian settings. Due to variation of available resources, diversity in dental education curriculum and requirements for accreditations, variable nature of COVID-19 impact on countries, relevant restrictions and compliance to public health messages amongst population, it was necessary to assess the impact on dental students in South Asian settings during the current COVID-19 pandemic.

Studies focusing on the impact of COVID-19 on Bangladeshi students, specifically, who were pursuing studies on health sciences including dentistry were very limited. However, it was important to assess their psychological impact not only due to the pandemic, but also due to the changed learning environment and clinical training. Therefore, we aimed to assess psychological distress, fear of COVID-19 and coping amongst Bangladeshi dental students and identify factors associated with those issues. Specifically, we intended to examine the extent of psychological distress, fear of COVID-19 and coping amongst them, and intended to identify the high-risk groups of individuals based on the identified factors utilizing validated study tools, so that future interventions can be targeted for such a cohort of dental students in Bangladesh. 

## 2. Materials and Methods

### 2.1. Study Design

A cross-sectional study was conducted from October to November 2021 where students of two different private Dental colleges participated via online platform.

### 2.2. Study Sites

Two large private Dental Colleges in the capital city of Bangladesh were selected as the study sites. Both sites had both teaching and clinical training facilities besides outdoor services. The first site had 444 students with 130 patients used to attending the hospital daily; the second site had 318 students with daily visit of 60 patients.

### 2.3. Study Population

Current students of those two study sites from first to fourth academic years were eligible for this study. Because of the inaccuracy of the responses, any study participant who took less than 1 min to complete the questionnaire was omitted from the analyses.

### 2.4. Sampling

The Snowball sampling technique was used for collecting data. Once a participant filled up the online questionnaire, he/she was requested to forward the survey link to his/her personal/professional networks. Sample size was calculated using Open Epi. Considering total students of 759 from two study sites, expected frequency of psychological distress as 70% based on the previous study in Bangladesh [8], 95% confidence intervals and 80% power, the estimated minimum sample size was 227.

### 2.5. Study Questionnaire

A structured survey questionnaire was used for data collection in this study and was adopted from an Australian and a global study led by the lead author of this study (MAR) [7,27]. Google forms were used to create the survey. The first section of the study questionnaire collected sociodemographic data as well as information on physical symptoms of COVID-19, history of contacts with COVID-19 cases, self-reported comorbidities, behavioral risk factors, health service utilization in the last four weeks including type of service providers and access to mental health resources. Psychological impact was assessed by the Kessler Psychological Distress Scale (K-10) [28], fear was assessed by the Fear of COVID-19 scale (FCV-19S) [29] and coping strategies were assessed by the Brief Resilient Coping Scale (BRCS) [30]. The details of each tool, for which the validity and the reliability were tested in previous studies, were discussed in our earlier published studies [27,31]. The K-10, having ten items, was scored based on the responses using a 5-point Likert scale; the scoring was categorized into low (10–15), moderate (16–21), high (22–29) and very high (30–50). The FCV-19S, having seven items, was scored based on the responses using a 5-point Likert scale; the scoring was categorized into low (7–21) and high (22–35). The BRCS, having four items, was scored based on the responses using a 5-point Likert scale; the scoring was categorized into low (4–13), medium (14–16) and high (17–20) [27]. The entire study questionnaire had a total of 46 items, which did not require more than 10 min to complete by a study participant. A pretest of the questionnaire was performed on a selective group of participants and the necessary modification was done before the data collection.

### 2.6. Data Collection

The online link of the survey was emailed to all the students at both sites inviting them to participate. The volunteer nature of the study was highlighted. Data were collected during October to November, 2021. On the first screen, the plain language information statement (PLIS) and consent form were displayed. Only those who provided consent could proceed to the following screens. The following screens displayed the entire study questionnaire.

### 2.7. Data Analyses

Data were analyzed using IBM SPSS v. 25 (Armonk, NY: IBM Corp.). At first, descriptive analyses were conducted. Categorical variables were reported as proportions and continuous variables were reported as mean (±SD). In that way, levels of psychological distress, fear of COVID-19 and coping were reported. Then, inferential analyses were conducted to identify the factors associated with those outcomes. At first, chi-squared tests were conducted to determine existence of association and *p* < 0.05 was considered significant. Then, univariate logistic regression was conducted to determine the strength of association; odds ratios (ORs) and 95% confidence intervals (CIs) were reported. Finally, multivariate logistic regression was conducted to control potential confounders; adjusted ORs (AORs) and 95% CIs were reported. In addition, to examine the relationship amongst the study tools, Pearson correlation tests and multiple linear regression were used with *p* < 0.05.

### 2.8. Ethics

The study protocol was reviewed and approved by the Human Research Ethics Committee at one of the study sites (ref no: SDC/C7/2021/829). The survey was completely voluntary in nature and it was clarified in the PLIS, so that participants got the opportunity to have an informed choice to participate in the study. No identifying information including any personal sensitive information were collected. Responses were anonymous and non-identifiable data were handled only by the study investigators.

## 3. Results

A total of 327 Bangladeshi dental undergraduate students participated in the study. The majority (71.6%) of the students belonged to the age group of 19–23 years and were females (74.9%). The mean age (± SD) was 22.5(±1.7) years with the majority (62.4%) from clinical years (third and fourth year). Almost all of the participants (95.1%) reported financial dependence on their families and more than half (58.1%) reported that the pandemic negatively impacted on their financial situation; yet most of them (81.9%) were satisfied with their current social life.

About one in five participants (19.6%) were infected with COVID-19, although recent infection was reported as only 2.4%. More than one in ten participants (15%) reported that they were involved in direct or indirect care of their family or friends who were infected with COVID-19. Other characteristics of the study population are reported in Table 1.

Though more than half of the participants reported low levels of fear of COVID-19 (53.8%), most of them experienced moderate to high level of psychological distress (84.2%) with more than half (60.2%) being low resilient copers. (Table 2, Table 3 and Table 4).

### 3.1. Psychological Distress

Univariate analyses showed that perceived safety in living places, being third year clinical dental students, negative impact of COVID-19 over financial situation, perceived satisfaction with current social life, irritating experience related to use of social media and feeling positive about life were significantly associated with moderate to very high levels of psychological distress compared to their counterparts. After adjustment of potential confounders, those who were at the third year of their academic year (AOR 0.14, 95% CIs 0.04–0.55, *p* = 0.005) and who reported satisfaction with current social life (AOR 0.11, 95% CIs 0.03–0.33, *p* < 0.001) were less likely to report moderate to very high levels of psychological distress. On the other hand, those who reported negative impact of COVID-19 over financial situation (AOR 3.72, 95% CIs 1.28–10.8, *p* = 0.015), who were infected with COVID-19 both recently and in the past, who were unsure of the contact with COVID-19 cases were more likely to report moderate to very high levels of psychological distress. (Table 5).

### 3.2. Levels of Fear

Univariate analyses showed that being female and those who were living in joint families were more likely to report high levels of fear of COVID-19. On the contrary, being a student of second, third and fourth year, being a smoker and those who were medium to high resilient copers were more likely to report low levels of fear of COVID-19. After adjustment of the potential confounders, it was found that those who were at the third year (AOR 0.17, 95% CIs 0.08–0.39, *p* < 0.001) and fourth year (AOR 0.29, 95% CIs 0.12–0.71, *p* = 0.006) clinical students had low levels of fear of COVID-19 (Table 6).

### 3.3. Coping Strategies

Univariate analyses demonstrated that those who were living in a hostel and those who had high level of COVID-19-related fear were more likely to be low resilient copers. On the other hand, those who were quite often positive about life and those who used health care services to overcome COVID-19-related stress were more likely to be medium to high resilient copers. However, after adjustment of the potential confounders, it was revealed that those who were females (AOR 0.47, 95% CIs 0.24–0.93, *p* = 0.030), living in hostel (AOR 0.51, 95% CIs 0.29–0.89, *p*= 0.018) and who were quite often positive about life (AOR 3.67, 95% CIs 1.17–11.5, *p* = 0.026) were more likely to be low resilient copers; those who were quite often positive about life (AOR 3.67, 95% CIs 1.17–11.5, *p* = 0.026) and those who used health care service to overcome COVID-19-related stress (AOR 2.19, 95% CIs 1.15–4.17, *p* = 0.017) were more likely to be medium to high resilient copers (Table 7).

### 3.4. Correlation within the Study Tools

When the total scoring of the K-10 tool was compared with the total scoring of the FCV-19S and BRCS tools, it was found that the psychological distress significantly predicted the fear of COVID-19 (r = 0.159, *p* < 0.01), but not the coping (r = −0.100, *p* > 0.05). Similarly, multiple linear regression also showed that the scoring of K-10 significantly predicted the scoring of FCV-19S (r = 0.258, *p* < 0.01), but not the scoring of BRCS (r = −0.305, *p* > 0.05) [F(2, 324) = 5.544, *p* < 0.01, R^2^ = 0.033].

## 4. Discussion

This is one of the very few studies conducted in Bangladesh amongst dental students about their mental health impacts during the current COVID-19 pandemic. Levels of psychological distress, fear and attempt to overcome the impact of ongoing pandemic was assessed; factors associated with those issues were also identified. Medical and dental education are considered highly stressful globally, because students experience higher levels of anxiety, stress and depression in comparison to students studying other subjects [32,33,34]. After the emergence of the COVID-19 pandemic, dental education was affected significantly owing to the need for reducing in-person contacts and enforcing social distancing in communities. Although several studies were conducted regarding psychological impact and fear of COVID-19 among the general population and medical students, this cross-sectional study was the first ever carried out in Bangladesh among dental students to assess the severity and to identify factors associated with psychological distress, levels of fear and coping strategies during the COVID-19 pandemic.

In this study, most of the dental students had moderate to high level of psychological distress (84.1%). That level was significantly higher than the medical students (65.9%) [35], general students (18.1%) [36] and general population (30.1%) [37] in Bangladesh during the COVID-19 pandemic. This might be due to increased risk of exposure of COVID-19 among dental students. Moreover, prior evidence indicated that dental education generated more stress and burnout than medical educations [38], due to more interactive involvement with patients during theoretical and clinical courses [39]. In this study, third year clinical students were more prone to having psychological distress due to COVID-19. A similar finding was reported in other studies, where clinical years had moderate to high levels of stress [40]. Another study by AL-Sowygh et al. showed that third year students had more stress due to performance pressure during clinical examination [22,41]. In this study, those who were infected with COVID-19 and who were unsure about the direct or indirect contact of COVID-19 cases were more prone to developing psychological distress. A similar finding was reported in the study conducted among the Australian population [27]. Those respondents in this study who reported a negative impact on their financial situations tended to have moderate to very high levels of psychological distress. As most of the dental students who took part in this study were fully dependent on their families, the negative financial impact could have hampered their academic progress.

Low levels of fear were reported amongst dental students in this study. That finding was in contrast to the finding from another study conducted in Bangladesh, which reported higher levels of fear amongst frontline or essential service workers [8]. Nevertheless, study findings from this study were consistent with the findings of another study where COVID-19-related fear was low among frontline health care workers. Similarly, low levels of fear among the doctors was observed in another study [42]. This might be due to increased engagement with the patients with a higher risk of exposure to COVID-19 and the availability of the protective gear during the time of data collection in Bangladesh. In this study, female dental students had higher levels of fear. Similar trends were observed among female dental and medical students and general population conducted elsewhere in Bangladesh [35,42,43,44,45]. This might be due to their inherent caregivers’ roles both in profession and families, hormonal changes, and expression of emotions, which could have contributed to the increased intensity of fear of COVID-19. In this study, third and fourth year clinical dental students had low levels of fear which was similar to a global study where doctors demonstrated lower levels of fear [7]. Medium to high resilient copers were more likely to have low level of fears in this study, which could be explained by the inherent capacity of high resilient copers to manage their fear, emotion, and stress more positively than the low resilient copers. Study participants who used healthcare services to combat COVID-19-related stress were more likely to be medium to high resilient copers. Similar findings were also reported in an earlier study, where visiting healthcare providers in persons was associated with high level of coping during the COVID-19 pandemic [7].

On the other hand, female students and students living in hostels tended to be low resilient copers. Students who had been living in hostels could have been dealing with a variety of concerns such as financial difficulties, home sickness, concerns on the safety of parents and relatives, change in sleeping and eating habits, and issues adjusting to their new surroundings, all of which probably made them more susceptible to psychological distress, hence low coping. Overall, this study identified that study participants were more low resilient copers, which could be due to high female respondents in this study. Although literature suggests that masculinity can explain part of the gender differences for stress and coping [46], in order to properly analyze these concerns, further research and study need to be conducted. In addition, further research could examine the link between coping and resources available for stress management in Bangladesh.

This study had few limitations. It was conducted among the students of two private dental colleges situated in Dhaka, Bangladesh, hence findings could not be generalized for all the dental students of Bangladesh. This was an online-based study, therefore the students who were only active online and had better internet connection were more likely to respond to this study. The inherent limitations of a cross-sectional study design could also not be ignored, which limited the ability for causal inference regarding the identified factors associated with psychological distress, fear and coping in this study. In addition, distressed students were more likely to respond in this study, which might have resulted in selection bias. On the other hand, dental students had different sorts of assessments and examination all the year round, so it could happen that the students who felt overwhelmed with their studies or clinical loads did not have time to respond to the survey. However, considering the ongoing pandemic crisis, it was inevitable to collect data online because of restriction of movement and social distancing. Nevertheless, this study was the first of its kind in Bangladesh to reveal the psychological distress, fear, and coping strategies of dental undergraduate students in Bangladesh.

Based on the findings from this study, few initiatives could be considered to support psychological wellbeing of dental students in Bangladesh. Counselling services should be incorporated into the dental institutes, where both staff and students would get access to resources and professionals during the crisis periods including such pandemic situations. Those services could be supported by the local institutes or Government. Trainings on pandemic and disaster preparedness should be incorporated as part of dental curriculum. Training on the use of personal protective equipment should be made mandatory for the third year dental students, where they commence their clinical placements, which would reduce distress and fear during such pandemic situations. Hybrid training models including both face-to-face and online components could be introduced incorporating theoretical, practical and clinical components, so that disruptions on learning could be minimized during any crisis period if the delivery options switched to online only. Finally, a financial support scheme should be considered for the students affected financially during the pandemic period. Easy student loan schemes could be considered from the institutes or Government.

## 5. Conclusions

This study identified that most of the dental students experienced moderate to very high levels of psychological distress while half of them had low levels of fear of COVID-19 with most of them being low resilient copers. The factors identified in this study should be considered in addressing mental health impacts of dental students in Bangladesh. Developing policies and support strategies for addressing health and wellbeing of dental students is imperative besides the core support for academic and clinical skills development. Future studies could focus on stakeholders and students of both public and private dental institutions in Bangladesh about the specific support strategies for psychological wellbeing during and post-pandemic.

## Figures and Tables

**Table 1 ijerph-19-00176-t001:** Characteristics of study population.

Characteristics	Total, *n* (%)
Total study participants	327
Age (in years)	327
Mean (±SD)	22.5 (1.7)
Age groups	327
19–23 years	234 (71.6)
24–28 years	93 (28.4)
Gender	327
Male	82 (25.1)
Female	245 (74.9)
Marital status	327
Married	48 (14.7)
Unmarried	278 (85.0)
Divorced	1 (0.3)
Family types	327
Nuclear family	266 (81.3)
Joint family	61 (18.7)
Living status	327
Live without family members	11 (3.4)
Live with family members	151 (46.2)
Live in own house	7 (2.1)
Live in shared house	16 (4.9)
Live in hostel	142 (43.4)
Perceived safety of living place in relation to COVID-19	327
Unsafe	50 (15.3)
Safe	277 (84.7)
Year of Dental education	327
1st year	68 (20.8)
2nd year	55 (16.8)
3rd year	98 (30.0)
4th year	106 (32.4)
Financial contribution to family	327
Fully dependent on family	311 (95.1)
Part of earning goes to family	16 (4.9)
COVID-19 impacted financial situation	327
No impact	82 (25.1)
Yes, impacted positively	54 (16.5)
Yes, impacted negatively	190 (58.1)
Perceived current social life	327
Dissatisfied	59 (18.0)
Satisfied	268 (81.9)
Co-morbidities	327
No	284 (86.9)
Diabetes	3 (0.9)
Hypertension	7 (2.1)
Tuberculosis	1 (0.3)
Chronic kidney disease	0 (0)
Lung disease	8 (2.4)
Carcinoma	0 (0)
Others	18 (5.5)
Smoking	327
Never smoker	311 (95.1)
Ever smoker (Daily/Non-daily/Ex)	16 (4.9)
Infected with COVID-19	327
No	227 (69.4)
Yes	64 (19.6)
Don’t know	36 (11)
Number of times infected with COVID-19	64
Mean (±SD)	1.14 (±0.393)
Infected with COVID-19 in the last 14 days	327
No	319 (97.6)
Yes	8 (2.4)
Experienced symptoms of COVID-19 in the last 14 days	327
No	259 (79.2)
Yes	46 (14.1)
May be	22 (6.7)
Contact (indirect/direct) with COVID-19 cases	327
No	278 (85.0)
Unsure	31 (9.5)
Yes	18 (5.5)
Activities during lockdown (multiple responses)	327
Reading books	4 (1.2)
Watching movies	4 (1.2)
Doing household chores	14 (4.3)
Listening to music	0 (0)
Engaging in social media	16 (4.9)
Cooking	8 (2.4)
Studying	31 (9.5)
Gardening	3 (.9)
Others	6 (1.8)
Experience related to the use of social media	327
Do not use	11 (3.4)
Does not affect	109 (33.3)
Find it irritating	207 (63.3)
Feel positive about life	327
Never	25 (7.6)
Quite often	150 (45.9)
Always positive	152 (46.5)
Faced difficulties in adopting distance learning	327
No	47 (14.4)
Yes	280 (85.6)
Healthcare service use to overcome COVID-19 related stress in the last 6 months	327
No	243 (74.3)
Yes	84 (25.7)
Type of healthcare service used to overcome COVID-19 related stress in the last 6 months	74
Consulted a GP	35 (10.7)
Consulted a Psychologist	19 (5.8)
Consulted a Psychiatrist	16 (4.9)
Others	4 (1.2)

**Table 2 ijerph-19-00176-t002:** Level of psychological distress among the study participants.

K-10 Items	Total, *n* (%)
About how often did you feel tired out for no good reason?	327
None	59 (18.0)
A little of the time	54 (16.5)
Some of the time	119 (36.4)
Most of the time	80 (24.5)
All of the time	15 (4.6)
About how often did you feel nervous?	327
None	51 (15.6)
A little of the time	77 (23.5)
Some of the time	111 (33.9)
Most of the time	75 (22.9)
All of the time	13 (4.0)
About how often did you feel so nervous that nothing could calm you down?	327
None	119 (36.4)
A little of the time	82 (25.1)
Some of the time	75 (22.9)
Most of the time	43 (13.1)
All of the time	8 (2.4)
About how often did you feel hopeless?	327
None	73 (22.3)
A little of the time	77 (23.5)
Some of the time	81 (24.8)
Most of the time	73 (22.3)
All of the time	23 (7.0)
About how often did you feel restless or fidgety?	327
None	70 (21.4)
A little of the time	82 (25.1)
Some of the time	95 (29.1)
Most of the time	67 (20.5)
All of the time	13 (4.0)
About how often did you feel so restless you could not sit still?	327
None	122 (37.3)
A little of the time	89 (27.2)
Some of the time	69 (21.1)
Most of the time	39 (11.9)
All of the time	8 (2.4)
About how often did you feel so depressed?	327
None	60 (20.2)
A little of the time	71 (21.7)
Some of the time	88(26.9)
Most of the time	75 (22.9)
All of the time	27 (8.3)
About how often did you feel that everything was an effort?	327
None	60 (18.3)
A little of the time	53 (16.2)
Some of the time	130 (39.8)
Most of the time	61 (18.7)
All of the time	23 (7.0)
About how often did you feel so sad that nothing could cheer you up?	327
None	84 (25.7)
A little of the time	82 (25.1)
Some of the time	89 (27.2)
Most of the time	61 (18.7)
All of the time	11 (3.4)
About how often did you feel worthless?	327
None	108 (33.0)
A little of the time	69 (21.1)
Some of the time	85 (26.0)
Most of the time	45 (13.8)
All of the time	20 (6.1)
K10 score (total)	327
Mean (±SD)	25.7 (9.1)
Level of psychological distress (K10 categories)	327
Low (score 10–15)	52 (15.9)
Moderate (score 16–21)	65 (19.9)
High (score 22–29)	97 (29.7)
Very high (score 30–50)	113 (34.6)

**Table 3 ijerph-19-00176-t003:** Level of fear of COVID-19 among the study participants.

FCV-19S Items	Total, *n* (%)
I am most afraid of COVID-19	327
Strongly disagree	27 (8.3)
Disagree	65 (19.9)
Neither agree nor disagree	92 (28.1)
Agree	127 (38.8)
Strongly agree	16 (4.9)
It makes me uncomfortable to think about COVID-19	327
Strongly disagree	15 (4.6)
Disagree	71 (21.7)
Neither agree nor disagree	77 (23.5)
Agree	147 (45.0)
Strongly agree	17 (5.2)
My hands become clammy when I think about COVID-19	327
Strongly disagree	39 (11.9)
Disagree	128 (39.1)]
Neither agree nor disagree	72 (22.0)
Agree	83 (25.4)
Strongly agree	5 (1.5)
I am afraid of losing my life because of COVID-19	327
Strongly disagree	30 (9.2)
Disagree	87 (26.6)
Neither agree nor disagree	70 (21.4)
Agree	118 (36.1)
Strongly agree	22 (6.7)
When watching news and stories about COVID-19 on social media, I become nervous or anxious	327
Strongly disagree	17 (5.2)
Disagree	55 (16.8)
Neither agree nor disagree	70 (21.4)
Agree	168 (51.4)
Strongly agree	17 (5.2)
I cannot sleep because I’m worrying about getting COVID-19	327
Strongly disagree	55 (16.8)
Disagree	153 (46.8)
Neither agree nor disagree	79 (24.2)
Agree	37 (11.3)
Strongly agree	3 (0.9)
My heart races or palpitates when I think about getting COVID-19	327
Strongly disagree	46 (14.1)
Disagree	113 (34.6)
Neither agree nor disagree	79 (24.2)
Agree	87 (26.6)
Strongly agree	2 (0.6)
FCV-19S score (total)	327
Mean (±SD)	20.4 (5.4)
Level of fear of COVID-19 (FCV-19S categories)	327
Low (score 7–21)	176 (53.8)
High (score 22–35)	151 (46.2)

**Table 4 ijerph-19-00176-t004:** Coping during the COVID-19 pandemic among the study participants.

BRCS Items	Total, *n* (%)
I look for creative ways to alter difficult situations	327
Does not describe me at all	16 (4.9)
Does not describe me	21 (6.4)
Neutral	186 (56.9)
Describes me	82 (25.1)
Describes me very well	22 (6.7)
Regardless of what happens to me, I believe I can control my reaction to it	327
Does not describe me at all	16 (4.9)
Does not describe me	27 (8.3)
Neutral	180 (55.0)
Describes me	79 (24.2)
Describes me very well	25 (7.6)
I believe I can grow in positive ways by dealing with difficult situations	327
Does not describe me at all	9 (2.8)
Does not describe me	19 (5.8)
Neutral	154 (47.1)
Describes me	112 (34.3)
Describes me very well	33 (10.1)
I actively look for ways to replace the losses I encounter in life	327
Does not describe me at all	13 (4.0)
Does not describe me	24 (7.3)
Neutral	190 (58.1)
Describes me	80 (24.5)
Describes me very well	20 (6.1)
BRCS score (total)	327
Mean (±SD)	13.1 (2.6)
Level of coping (BRCS categories)	327
Low resilient copers (score 4–13)	197 (60.2)
Medium resilient copers (score 14–16)	102 (3.2)
High resilient copers (score 17–20)	28 (8.6)

**Table 5 ijerph-19-00176-t005:** Predictors for high psychological distress among the study population (based on K10 score).

Characteristics	Low (Score 10–15)		Moderate to Very High (Score 16–50)		Unadjusted Analyses			Adjusted Analyses		
	*n*	%	*n*	%	*p*	ORs	95% CIs	*p*	AORs	95% CIs
Age groups	52		275							
19–23 years	38	16.2	196	83.8		1			1	
24–28 years	14	15.1	79	84.9	0.791	1.09	0.56–2.13	0.188	0.43	0.12–1.51
Gender	52		275							
Male	18	22	64	78		1			1	
Female	34	13.9	211	86.1	0.086	1.75	0.92–3.30	0.130	2.25	0.79–6.45
Marital status	52		275							
Married	8	16.7	40	83.3		1			1	
Unmarried	44	15.8	234	84.2	0.883	1.06	0.47–2.43	0.465	0.63	0.18–2.19
Family types	52		275							
Nuclear family	41	15.4	225	84.6		1			1	
Joint family	11	18	50	82	0.614	0.83	0.40–2.72	0.825	0.88	0.29–2.65
Living status	52		275							
Live without family members	1	9.1	10	90.9		1			1	
Live with family members	23	15.2	128	84.8	0.585	1.80	0.22–14.7	0.299	4.26	0.28–65.4
Live in own house	2	28.6	5	71.4	0.356	0.45	0.08–2.46	0.412	0.38	0.04–3.88
Live in shared house	4	25	12	75	0.319	0.54	0.16–1.82	0.609	1.58	0.28–9.01
Live in hostel	22	15.5	120	84.5	0.951	0.98	0.52–1.85	0.802	1.14	0.41–3.15
Perceived safety of living place in relation to COVID-19	52		275							
Unsafe	3	6	47	94		1			1	
Safe	49	17.7	228	82.3	** *0.049* **	** *0.30* **	** *0.09–0.99* **	0.970	1.03	0.17–6.21
Year of Dental education	52		275							
1st year	7	10.3	61	89.7		1			1	
2nd year	5	9.1	50	90.9	0.823	1.15	0.34–3.84	0.718	0.74	0.15–3.74
3rd year	29	29.6	69	70.4	** *0.004* **	** *0.27* **	** *0.11–0.67* **	** *0.005* **	** *0.14* **	** *0.04–0.55* **
4th year	11	10.4	95	89.6	0.986	0.99	0.36–2.70	0.858	0.87	0.19–4.07
Financial contribution to family	52		275							
Fully dependent on family	49	15.8	262	84.2		1			1	
Part of earning goes to family	3	18.8	13	81.3	0.750	0.81	0.22–2.95	0.953	1.07	0.13–8.62
COVID-19 impacted financial situation	52		275							
No impact	26	31.7	56	68.3		1			1	
Yes, impacted positively	9	16.7	45	83.3	0.053	2.32	0.99–5.45	0.288	2.01	0.56–7.24
Yes, impacted negatively	17	8.9	173	91.1	** *0.000* **	** *4.72* **	** *2.39–9.34* **	** *0.015* **	** *3.72* **	** *1.28–10.8* **
Perceived current social life	52		275							
Dissatisfied	10	5.6	170	94.4		1			1	
Satisfied	42	28.6	105	71.4	** *0.000* **	** *0.15* **	** *0.07–0.31* **	** *0.000* **	** *0.11* **	** *0.03–0.33* **
Smoking	52		275							
Never smoker	51	16.4	260	83.6		1			1	
Ever smoker (Daily/Non-daily/Ex)	1	6.3	15	93.8	0.301	2.94	0.38–22.8	0.232	6.22	0.31–125
Infected with COVID-19	52		275							
No	40	17.6	187	82.4		1			1	
Yes	8	12.5	56	87.5	0.215	1.36	0.84–2.22	** *0.030* **	** *2.52* **	** *1.10–5.78* **
Infected with COVID-19 in the last 14 days	52		275							
No	51	16	268	84		1			1	
Yes	1	12.5	7	87.5	0.791	1.33	0.16–11.1	** *0.022* **	** *62.7* **	** *1.81–2175* **
Experienced symptoms of COVID-19 in the last 14 days	52		275							
No	44	17	215	83		1			1	
Yes	6	13	40	87	0.384	1.23	0.78–1.93	0.104	0.48	0.20–1.17
Contact (indirect/direct) with COVID-19 cases	52		275							
No	45	16.2	233	83.8		1			1	
Unsure	3	9.7	28	90.3	0.349	1.80	0.53–6.18	** *0.030* **	** *8.38* **	** *1.23–56.9* **
Yes	4	22.2	14	77.8	0.507	0.68	0.21–2.15	0.279	0.32	0.04–2.51
Experience related to the use of social media	52		275							
Do not use	4	36.4	7	63.6		1			1	
Does not affect	32	29.4	77	70.6	0.630	1.38	0.38–5.02	0.870	1.16	0.19–7.16
Find it irritating	16	7.7	191	92.3	** *0.005* **	** *6.82* **	** *1.80–25.8* **	0.158	3.88	0.59–25.5
Feel positive about life	52		275							
Never	1	4	24	96		1			1	
Quite often	3	2	147	98	0.544	2.040	0.20–20.4	0.174	7.16	0.42–122
Always positive	48	31.6	104	68.4	** *0.020* **	** *0.09* **	** *0.01–0.69* **	0.292	0.26	0.02–3.17
Faced difficulties in adopting distance learning	52		275							
No	12	25.5	35	74.5		1			1	
Yes	40	14.3	240	85.7	0.055	2.06	0.99–4.30	0.060	2.95	0.96–9.12
Level of fear of COVID-19 (FCV-19S categories)	52		275							
Low (score 7–21)	26	14.8	150	85.2		1			1	
High (score 22–35)	26	17.2	125	82.8	0.547	0.83	0.46–1.51	0.117	0.45	0.16–1.22
Level of coping (BRCS categories)	52		275							
Low resilient copers (score 4–13)	34	17.3	163	82.7		1			1	
Medium to high resilient copers (score 14–20)	18	13.8	112	86.2	0.410	1.30	0.70–2.41	0.109	0.44	0.16–1.20
Healthcare service use to overcome COVID-19 related stress in the last 6 months	52		275							
No	39	16	204	84		1			1	
Yes	13	15.5	71	84.5	0.901	0.96	0.48–1.90	0.267	0.52	0.16–1.65
Type of healthcare service used to overcome COVID-19 related stress in the last 6 months										
Consulted a GP	6	17.1	29	82.9	0.928	1.06	0.32–3.51	0.700	1.61	0.14–18.3
Consulted a Psychologist	5	26.3	14	73.7	0.252	0.48	0.13–1.69	0.957	0.93	0.08–11.2
Consulted a Psychiatrist	1	6.3	15	93.8	0.208	3.91	0.47–32.7	0.299	5.00	0.24–104
Others	1	25	3	75	0.690	0.62	0.06–6.49	NA	NA	NA

Adjusted for: age, gender, marital status, family types, living status, perceived safety of living, year of dental education, financial contribution to family, financial impact, perceived social life, smoking, infected with COVID-19 ever or in the last 14 days, COVID symptoms, contacts with COVID cases, experience related to social media use, feel positive about life, adopting distance learning, levels of fear and coping, healthcare service use and types. Bold Italics indicated statistical significance in the table.

**Table 6 ijerph-19-00176-t006:** Predictors for fear of COVID-19 among the study population (based on FCV-19S score).

Characteristics	Low (Score 7–21)		High (Score 22–35)		Unadjusted Analyses			Adjusted Analyses		
	n	%	*n*	%	*p*	ORs	95% CIs	*p*	AORs	95% CIs
Age groups	176		151							
19–23 years	118	50.4	116	49.6		1			1	
24–28 years	58	62.4	35	37.6	0.052	0.61	0.38–1.00	0.717	0.87	0.42–1.81
Gender	176		151							
Male	57	69.5	25	30.5		1			1	
Female	119	48.6	126	51.4	** *0.001* **	** *2.41* **	** *1.42–4.11* **	0.120	1.71	0.87–3.37
Marital status	176		151							
Married	26	54.2	22	45.8		1			1	
Unmarried	150	54	128	46	0.979	1.01	0.55–1.86	0.766	0.89	0.41–1.93
Family types	176		151							
Nuclear family	151	56.8	115	43.2		1			1	
Joint family	25	41	36	59	** *0.027* **	** *1.89* **	** *1.07–3.33* **	0.054	1.94	0.99–3.81
Living status	176		151							
Live without family members	6	54.5	5	45.5		1			1	
Live with family members	78	51.7	73	48.3	0.853	0.89	0.26–3.04	0.605	0.67	0.14–3.11
Live in own house	4	57.1	3	42.9	0.777	0.80	0.17–3.70	0.790	0.79	0.13–4.61
Live in shared house	11	68.8	5	31.3	0.200	0.49	0.16–1.47	0.107	0.31	0.08–1.29
Live in hostel	77	54.2	65	45.8	0.660	0.90	0.57–1.43	0.181	0.68	0.39–1.20
Perceived safety of living place in relation to COVID-19	176		151							
Unsafe	25	50	25	50		1			1	
Safe	151	54.5	126	45.5	0.556	0.83	0.46–1.52	0.251	0.64	0.30–1.37
Year of Dental education	176		151							
1st year	21	30.9	47	69.1		1			1	
2nd year	28	50.9	27	49.1	** *0.025* **	** *0.43* **	** *0.21–0.90* **	0.064	0.45	0.20–1.05
3rd year	63	64.3	35	35.7	** *0.000* **	** *0.25* **	** *0.13–0.48* **	** *0.000* **	** *0.17* **	** *0.08–0.39* **
4th year	64	60.4	42	39.6	** *0.000* **	** *0.29* **	** *0.15–0.56* **	** *0.006* **	** *0.29* **	** *0.12–0.71* **
Financial contribution to family	176		151							
Fully dependent on family	164	52.7	147	47.3		1			1	
Part of earning goes to family	12	75	4	25	0.093	0.37	0.12–1.18	0.225	0.45	0.12–1.64
COVID-19 impacted financial situation	176		151							
No impact	49	59.8	33	40.2		1			1	
Yes, impacted positively	31	57.4	23	42.6	0.785	1.10	0.55–2.21	0.764	0.88	0.38–2.05
Yes, impacted negatively	96	50.5	94	49.5	0.162	1.45	0.86–2.46	0.116	1.67	0.88–3.18
Perceived current social life	176		151							
Dissatisfied	94	52.2	86	47.8		1			1	
Satisfied	82	55.8	65	44.2	0.521	0.87	0.56–1.34	0.235	0.71	0.40–1.25
Smoking	176		151							
Never smoker	162	52.1	149	47.9		1			1	
Ever smoker (Daily/Non-daily/Ex)	14	87.5	2	12.5	** *0.015* **	** *0.16* **	** *0.03–0.69* **	0.067	0.20	0.04–1.12
Infected with COVID-19	176		151							
No	121	53.3	106	46.7		1			1	
Yes	36	56.3	28	43.8	0.896	0.98	0.71–1.35	0.639	1.10	0.74–1.63
Infected with COVID-19 in the last 14 days	176		151							
No	173	54.2	146	45.8		1			1	
Yes	3	37.5	5	62.5	0.357	1.97	0.46–8.40	0.513	1.79	0.31–10.3
Experienced symptoms of COVID-19 in the last 14 days	176		151							
No	139	53.7	120	46.3		1			1	
Yes	26	56.5	20	43.5	0.798	0.96	0.71–1.30	0.128	0.69	0.43–1.11
Contact (indirect/direct) with COVID-19 cases	176		151							
No	152	54.7	126	45.3		1			1	
Unsure	17	54.8	14	45.2	0.986	0.99	0.47–2.09	0.524	1.37	0.52–3.57
Yes	7	38.9	11	61.1	0.199	1.90	0.71–5.03	0.155	2.48	0.71–8.71
Experience related to the use of social media	176		151							
Do not use	5	45.5	6	54.5		1			1	
Does not affect	69	63.3	40	36.7	0.254	0.48	0.14–1.68	0.268	0.43	0.09–1.93
Find it irritating	102	49.3	105	50.7	0.805	0.860	0.25–2.90	0.822	0.84	0.19–3.70
Feel positive about life	176		151							
Never	12	48	13	52		1			1	
Quite often	93	62	57	38	0.190	0.57	0.24–1.33	0.155	0.47	0.17–1.33
Always positive	71	46.7	81	53.3	0.905	1.05	0.45–2.46	0.485	1.46	0.50–4.25
Faced difficulties in adopting distance learning	176		151							
No	26	55.3	21	44.7		1			1	
Yes	150	53.6	130	46.4	0.824	1.07	0.58–2.00	0.646	0.84	0.39–1.80
Level of psychological distress (K10 categories)	176		151							
Low (score 10–15)	26	50	26	50		1			1	
Moderate to Very High (score 16–50)	150	54.5	125	45.5	0.547	0.83	0.46–1.51	0.190	0.58	0.26–1.31
Level of coping (BRCS categories)	176		151							
Low resilient copers (score 4–13)	97	49.2	100	50.8		1			1	
Medium to high resilient copers (score 14–20)	79	60.8	51	39.2	** *0.041* **	** *0.63* **	** *0.40–0.98* **	0.140	0.67	0.39–1.14
Healthcare service use to overcome COVID-19 related stress in the last 6 months	176		151							
No	133	54.7	110	45.3		1			1	
Yes	43	51.2	41	48.8	0.575	0.87	0.53–1.43	0.934	1.03	0.55–1.93
Type of healthcare service used to overcome COVID-19 related stress in the last 6 months										
Consulted a GP	18	51.4	17	48.6	0.501	0.73	0.29–1.82	0.387	2.83	0.27–30.0
Consulted a Psychologist	7	36.8	12	63.2	0.293	1.78	0.61–5.19	0.190	5.14	0.44–59.5
Consulted a Psychiatrist	7	43.8	9	56.3	0.748	1.20	0.39–3.66	0.284	3.86	0.33–45.6
Others	3	75	1	25	0.281	0.28	0.03–2.83	NA	NA	NA

Adjusted for: age, gender, marital status, family types, living status, perceived safety of living, year of dental education, financial contribution to family, financial impact, perceived social life, smoking, infected with COVID-19 ever or in the last 14 days, COVID symptoms, contacts with COVID cases, experience related to social media use, feel positive about life, adopting distance learning, levels of psychological distress and coping, healthcare service use and types. Bold Italics indicated statistical significance in the table.

**Table 7 ijerph-19-00176-t007:** Predictors for coping among the study population (based on BRCS score).

Characteristics	Low (Score 4–13)		Medium to High (Score 14–20)		Unadjusted Analyses			Adjusted Analyses		
	*n*	%	*n*	%	*p*	ORs	95% CIs	*p*	AORs	95% CIs
Age groups	197		130							
19–23 years	146	62.4	88	37.6		1			1	
24–28 years	51	54.8	42	45.2	0.209	1.37	0.84–2.22	0.218	1.56	0.77–3.14
Gender	197		130							
Male	43	52.4	39	47.6		1			1	
Female	154	62.9	91	37.1	0.096	0.65	0.39–1.08	** *0.030* **	** *0.47* **	** *0.24–0.93* **
Marital status	197		130							
Married	26	54.2	22	45.8		1			1	
Unmarried	170	61.2	108	38.8	0.362	0.75	0.41–1.39	0.481	0.77	0.37–1.60
Family types	197		130							
Nuclear family	160	60.2	106	39.8		1			1	
Joint family	37	60.7	24	39.3	0.942	0.98	0.55–1.73	0.774	0.91	0.47–1.76
Living status	197		130							
Live without family members	9	81.8	2	18.2		1			1	
Live with family members	82	54.3	69	45.7	0.095	0.26	0.06–1.26	0.051	0.16	0.03–1.01
Live in own house	3	42.9	4	57.1	0.556	1.58	0.34–7.32	0.424	2.02	0.36–11.4
Live in shared house	9	56.3	7	43.8	0.882	0.92	0.33–2.61	0.977	1.02	0.31–3.33
Live in hostel	94	66.2	48	33.8	** *0.038* **	** *0.61* **	** *0.38–0.97* **	** *0.018* **	** *0.51* **	** *0.29–0.89* **
Perceived safety of living place in relation to COVID-19	197		130							
Unsafe	27	54	23	46		1			1	
Safe	170	61.4	107	38.6	0.328	0.74	0.40–1.36	0.227	0.64	0.31–1.32
Year of Dental education	197		130							
1st year	46	67.6	22	32.4		1			1	
2nd year	28	50.9	27	49.1	0.061	2.02	0.97–4.20	0.161	1.82	0.79–4.20
3rd year	61	62.2	37	37.8	0.475	1.27	0.66–2.43	0.663	0.84	0.39–1.83
4th year	62	58.5	44	41.5	0.226	1.48	0.78–2.81	0.493	0.74	0.31–1.75
Financial contribution to family	197		130							
Fully dependent on family	189	60.8	122	39.2		1			1	
Part of earning goes to family	8	50	8	50	0.394	1.55	0.57–4.24	0.893	1.08	0.34–3.49
COVID-19 impacted financial situation	197		130							
No impact	55	67.1	27	32.9		1			1	
Yes, impacted positively	35	64.8	19	35.2	0.785	1.11	0.54–2.28	0.974	1.01	0.44–2.34
Yes, impacted negatively	106	55.8	84	44.2	0.084	1.61	0.94–2.78	0.245	1.46	0.77–2.77
Perceived current social life	197		130							
Dissatisfied	104	57.8	76	42.2		1			1	
Satisfied	93	63.3	54	36.7	0.313	0.79	0.51–1.24	0.890	1.04	0.60–1.80
Smoking	197		130							
Never smoker	190	61.1	121	38.9		1			1	
Ever smoker (Daily/Non-daily/Ex)	7	43.8	9	56.3	0.174	2.02	0.73–5.56	0.462	1.62	0.45–5.82
Infected with COVID-19	197		130							
No	143	63	84	37		1			1	
Yes	35	54.7	29	45.3	0.140	1.27	0.92–1.76	0.093	1.39	0.95–2.04
Infected with COVID-19 in the last 14 days	197		130							
No	193	60.5	126	39.5		1			1	
Yes	4	50	4	50	0.552	1.53	0.38–6.24	0.389	2.19	0.37–12.9
Experienced symptoms of COVID-19 in the last 14 days	197		130							
No	154	59.5	105	40.5		1			1	
Yes	29	63	17	37	0.599	0.92	0.67–1.26	0.352	0.81	0.51–1.27
Contact (indirect/direct) with COVID-19 cases	197		130							
No	164	59	114	41		1			1	
Unsure	23	74.2	8	25.8	0.106	0.50	0.22–1.16	0.096	0.42	0.15–1.17
Yes	10	55.6	8	44.4	0.774	1.15	0.44–3.01	0.729	1.24	0.37–4.12
Experience related to the use of social media	197		130							
Do not use	7	63.6	4	36.4		1			1	
Does not affect	71	65.1	38	34.9	0.921	0.94	0.26–3.40	0.359	0.51	0.12–2.14
Find it irritating	119	57.5	88	42.5	0.688	1.29	0.37–4.56	0.815	0.85	0.21–3.45
Feel positive about life	197		130							
Never	19	76	6	24		1			1	
Quite often	80	53.3	70	46.7	** *0.040* **	** *2.77* **	** *1.05–7.33* **	** *0.026* **	** *3.67* **	** *1.17–11.5* **
Always positive	98	64.5	54	35.5	0.264	1.74	0.66–4.63	0.084	2.83	0.87–9.20
Faced difficulties in adopting distance learning	197		130							
No	34	72.3	13	27.7		1			1	
Yes	163	58.2	117	41.8	0.070	1.88	0.95–3.71	0.120	1.87	0.85–4.09
Level of psychological distress (K10 categories)	197		130							
Low (score 10–15)	34	65.4	18	34.6		1			1	
Moderate to Very High (score 16–50)	163	59.3	112	40.7	0.410	1.30	0.70–2.41	0.890	0.94	0.42–2.11
Level of fear of COVID-19 (FCV-19S categories)	197		130							
Low (score 7–21)	97	55.1	79	44.9		1			1	
High (score 22–35)	100	66.2	51	33.8	** *0.041* **	** *0.63* **	** *0.40–0.98* **	0.131	0.66	0.38–1.13
Healthcare service use to overcome COVID-19 related stress in the last 6 months	197		130							
No	137	56.4	106	43.6		1			1	
Yes	60	71.4	24	28.6	** *0.016* **	** *1.93* **	** *1.13–3.31* **	** *0.017* **	** *2.19* **	** *1.15–4.17* **
Type of healthcare service used to overcome COVID-19 related stress in the last 6 months										
Consulted a GP	24	68.6	11	31.4	0.421	1.53	0.54–4.29	0.793	1.38	0.13–14.8
Consulted a Psychologist	16	84.2	3	15.8	0.210	0.42	0.11–1.63	0.662	0.56	0.04–7.40
Consulted a Psychiatrist	11	68.8	5	31.3	0.668	1.30	0.39–4.38	0.808	1.36	0.11–16.6
Others	3	75	1	25	0.925	0.89	0.09–9.14	NA	NA	NA

Adjusted for: age, gender, marital status, family types, living status, perceived safety of living, year of dental education, financial contribution to family, financial impact, perceived social life, smoking, infected with COVID-19 ever or in the last 14 days, COVID symptoms, contacts with COVID cases, experience related to social media use, feel positive about life, adopting distance learning, levels of psychological distress and fear, healthcare service use and types. Bold Italics indicated statistical significance in the table.

## Data Availability

The data are available upon reasonable request from the corresponding author.

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
