# Peer review of "Psychological Distress among Bangladeshi Dental Students during the COVID-19 Pandemic"

_ijerph, 2021, doi:10.3390/ijerph19010176_

Round 1

Reviewer 1 Report

Thank you for the opportunity to review this paper. The study addresses an important topic - stress and health concerns for health professions students and the impact of COVID-19. This is a very important area for research. Introduction Good background information. Identified the changes made to the learning environment because of COVID-19. Briefly identified the increase in psychological stress experienced.

Line 123 Aim is included. Specific research question isn’t included. No hypothesis.

Line 120 – 121 Need to clearly outline how the study differs / contributes to the existing studies – reference 23 – 26. (e.g. like what is included in lines 254 – 258). This paragraph says that there are limited studies but the previous paragraph identifies four references that focus on the impact of COVID-19 on students. Clearly articulating what this study contributes will help the reader.

2.1 Study design Possibly explain if this time of year is when students have assessment task due, ie. it is a particularly stressful time of year? A limitation to surveying students during a stressful / demanding period in the academic year, students are less likely to complete a survey / have the time to complete a survey. You could potentially miss students who are feeling overwhelmed. Ie. they are the ones who are less likely to find time for a survey. Would the final year students be at risk of stress due to the impact of COVID-19.

2.5 Study Questionnaire I am aware that the study is linked to other published works. However, it would help the reader to include details surrounding the number of questions in total / the time taken to complete the survey. Especially considering the comment in Line 137 – 138. Ie. 1 minute duration were excluded. Consider moving the survey length up from line 165. Line 263 More detail about how dental students experience higher levels of stress might add value to this section. Do the findings from reference 36 relate to your findings? For example like what is written in line 264 – good. Line 280 Sentence structure.

Results Tables 5 – 7 are clear and well presented. Possible reconsider the univariate tables. A lot of data included – can the measures be reported as a single score? The scoring process isn’t included in the survey section / data analysis section. I’m assuming each item is being reported individually for a reason / as per the measures author’s instructions(?) Possible data suggestion – it could be beneficial for the reader to see a linear regression for the main variables (e.g. fear and stress).

Discussion Table 1 presents interesting information. Where there any significant / links between the activities completed during lockdown and coping styles? Further research could examine the link between coping and resources available for stress management.

Line 287 – a good link to female students and the caregiver role. This could be nicely linked to theories about gender and stress / interpersonal professions. This might be of interest.

Line 306 – need to include a paragraph explaining how the findings can be used to inform dental student training programs? How can students be supported in light of these findings? The results are explained however implications for practice aren’t clearly defined. Possible suggest support for the ‘high risks’ groups identified in your participant sample.

Line 312 – consider also that students feeling overwhelmed may not have time to complete a survey. Surveys carry a degree of bias because it relies on participants reflecting on their own behaviour. General comments Check sentence structure. Possible re-write. E.g. lines 54 – 59; 100 – 102; 106 – 108; 249 – 254.

Reference to relevant theories Coping strategies aren’t clearly defined in the introduction. Ie. why different coping strategies are linked to lower / higher psychological distress. Examples of studies/existing research are provided in the discussion. But there is limited reference to relevant theories for coping / stress (e.g. Lazarus and Folkman etc). This could be included in the introduction with information about which groups within the dental student population would be considered high risk. This is mentioned briefly (e.g. lines 73 – 80).

Author Response

We would like to sincerely thank you for the opportunity to revise our manuscript. We appreciate the feedback and constructive suggestions provided by the reviewers. 

Reviewer 2 Report

Thanks for recommending me as a reviewer. In this paper, the authors assessed the levels and identified factors associated with psychological distress, fear and coping experienced by dental undergraduate students in Bangladesh. If the authors complete the revision, the quality of the study will be further improved.

  1. The introduction section is well written. It may be helpful for readers to understand if the authors describe in more detail the trends of previous studies on the 'correlation between the COVID-19 pandemic and the psychological distress of college students' in the introduction section. A lot of previous studies have been accumulated on the relationship between the current COVID-19 pandemic and the psychological pain of college students.

2. line 126-129: "2.1. Study design" - In section "2.1. Study design", authors should add sampling.

3. line 149-158: Are the tests used in this study standardized? What is the cut-off score for each test? If the authors describe the tests more specifically in the Methods section, it may be helpful for readers to understand.

4. Authors need to describe in detail information on “Adjusted analyses” (ex. confounding variables) in the footnotes of Tables 5 and 6.

5. Authors should add further study limitations to the discussion section.

Author Response

(The authors gave the same response as above.)

Round 2

Reviewer 1 Report

Thank you for providing a response to the suggestions / feedback. The revisions are well-considered and prepared.